# Beneficial Interactive Effects Provided by an Arbuscular Mycorrhizal Fungi and Yeast on the Growth of *Oenothera picensis* Established on Cu Mine Tailings

**DOI:** 10.3390/plants12234012

**Published:** 2023-11-29

**Authors:** Rodrigo Pérez, Yasna Tapia, Mónica Antilén, Antonieta Ruiz, Paula Pimentel, Christian Santander, Humberto Aponte, Felipe González, Pablo Cornejo

**Affiliations:** 1Plant Stress Physiology Laboratory, Centro de Estudios Avanzados en Fruticultura (CEAF), Rengo 2940000, Chile; rodrigoestebanperez@gmail.com (R.P.); ppimentel@ceaf.cl (P.P.); 2Departamento de Ingeniería y Suelos, Universidad de Chile, Santiago 8820808, Chile; yasnatapiafernandez@uchile.cl; 3Departamento de Química Inorgánica, Pontificia Universidad Católica de Chile, Santiago 7820436, Chile; mantilen@uc.cl; 4Departamento de Ciencias Químicas y Recursos Naturales, Facultad de Ingeniería y Ciencias, Universidad de La Frontera, Avenida Francisco Salazar 01145, Temuco 4811230, Chile; maria.ruiz@ufrontera.cl (A.R.); c.santander01@ufromail.cl (C.S.); 5Grupo de Ingeniería Ambiental y Biotecnología, Facultad de Ciencias Ambientales y Centro EULA-Chile, Universidad de Concepción, Concepción 4070411, Chile; 6Laboratory of Soil Microbiology and Biogeochemistry, Institute of Agri-Food, Animal and Environmental Sciences (ICA3), Universidad de O’Higgins, San Fernando 3070000, Chile; humberto.aponte@uoh.cl; 7Centre of Systems Biology for Crop Protection (BioSav), Institute of Agri-Food, Animal and Environmental Sciences (ICA3), Universidad de O’Higgins, San Fernando 3070000, Chile; 8Programa de Doctorado en Ciencias Mención Biología Celular y Molecular Aplicada, Universidad de La Frontera, Temuco 4780000, Chile; f.gonzalez31@ufromail.cl; 9Escuela de Agronomía, Facultad de Ciencias Agronómicas y de los Alimentos, Pontificia Universidad Católica de Valparaíso, San Francisco S/N, La Palma, Quillota 2260000, Chile; 10Centro Regional de Investigación e Innovación para la Sostenibilidad de la Agricultura y los Territorios Rurales, CERES, La Palma, Quillota 2260000, Chile

**Keywords:** antioxidant response, arbuscular mycorrhizal fungi, copper, metal(loid) contamination, mine waste, organic amendments, phytostabilization

## Abstract

Phytoremediation, an environmentally friendly and sustainable approach for addressing Cu-contaminated environments, remains underutilized in mine tailings. Arbuscular mycorrhizal fungi (AMF) play a vital role in reducing Cu levels in plants through various mechanisms, including glomalin stabilization, immobilization within fungal structures, and enhancing plant tolerance to oxidative stress. Yeasts also contribute to plant growth and metal tolerance by producing phytohormones, solubilizing phosphates, generating exopolysaccharides, and facilitating AMF colonization. This study aimed to assess the impact of AMF and yeast inoculation on the growth and antioxidant response of *Oenothera picensis* plants growing in Cu mine tailings amended with compost. Plants were either non-inoculated (NY) or inoculated with *Meyerozyma guilliermondii* (MG), *Rhodotorula mucilaginosa* (RM), or a combination of both (MIX). Plants were also inoculated with *Claroideoglomus claroideum* (CC), while others remained non-AMF inoculated (NM). The results indicated significantly higher shoot biomass in the MG-NM treatment, showing a 3.4-fold increase compared to the NY-NM treatment. The MG-CC treatment exhibited the most substantial increase in root biomass, reaching 5-fold that in the NY-NM treatment. Co-inoculation of AMF and yeast influenced antioxidant activity, particularly catalase and ascorbate peroxidase. Furthermore, AMF and yeast inoculation individually led to a 2-fold decrease in total phenols in the roots. Yeast inoculation notably reduced non-enzymatic antioxidant activity in the ABTS and CUPRAC assays. Both AMF and yeast inoculation promoted the production of photosynthetic pigments, further emphasizing their importance in phytoremediation programs for mine tailings.

## 1. Introduction

In recent years, the environmental contamination resulting from heavy metal(loid) concentrations has significantly increased, presenting a global environmental threat [1]. Mining activities have led to the accumulation of an estimated 5 to 14 million tons of mine waste tailings in recent decades, giving rise to a range of issues associated with tailing management [2]. These tailings disrupt topsoil and vegetation, reduce soil biodiversity, and create harsh environmental conditions characterized by infertile soils and elevated concentrations of pollutants, particularly toxic heavy metal(loids), thereby restricting natural revegetation and soil restoration [3,4,5,6]. The chemical composition of mine tailings varies depending on the characteristics of the mined rock and may include several primary components, such as Si, S, Fe, Al, Ca, Cu, Mg, K, Na, and Mn [5,7,8]. Mine tailings, originating from minerals, have low levels of organic matter and nutrients and lack the organic structure and support required for microorganisms [4,8,9]. Furthermore, they exhibit high concentrations of heavy metal(loid)s and often have acidic pH levels, which promotes erosion and causes detrimental effects to plants and soil microorganisms [5,6,10].

Reestablishing native or pioneer plants is considered an essential step in restoring affected sites. To promote sustainable mine tailing management, the use of plants in bioremediation, or phytoremediation, is recommended. This approach helps to alleviate the toxicity of heavy metal(loid)s, maintains plant cover to stabilize the surface of tailings, and prevents further erosion caused by wind and water [6,11,12]. Several physicochemical conditions of mine tailings impose significant limitations on in situ plant establishment. To overcome these limitations, organic amendments and plant growth-promoting (PGP) microorganisms have been applied to facilitate and enhance plant establishment and growth [3,13,14,15,16]. To improve copper (Cu) phytoremediation, metallophytes tolerant to high Cu concentrations are required. Cu-adapted metallophytes have developed diverse strategies to exclude, accumulate, and stabilize the Cu in their tissues or beyond the root [17,18]. In addition, the presence of other beneficial traits that increase tolerance to extremely low or high pH, severe drought, salinity, and low contents of organic matter and mineral nutrients must be considered since such conditions are commonly present in mine tailings [19,20]. Regarding the above, *Oenothera picensis* is a biennial plant commonly present in degraded soils in Central Chile [21], being able to grow in soils contaminated with Cu mine waste [18,22]. Furthermore, *O. picensis* has shown high capability for Cu accumulation in root tissues [13], high biomass production, and drought resistance, therefore emerging as a potential candidate to be used in the phytoremediation of Cu-contaminated environments [17,23,24,25]. Recently, Pérez et al. [16] reported that *O. picensis* plants can grow on Cu mine tailings, and they found a significant positive effect of up to 2-fold on biomass production when compost and arbuscular mycorrhizal fungi (AMF) were used in combination. Furthermore, the combined effect increased the mycorrhizal structures in the root and the nutritional characteristics in *O. picensis*; meanwhile, a decrease in the Cu availability in mine tailings was observed.

Previous studies have reported that colonization by AMF can result in a reduction in the overall Cu concentration in plants while also increasing Cu stabilization through various mechanisms, including glomalin production, immobilization by the fungal walls of hyphae and spores, and storage of Cu within resistant spores [16,25,26,27]. However, this symbiosis also exerts effects modulating the molecular mechanisms of plants to cope with the stress produced by toxic elements such as Cu, mainly regarding the response of antioxidant enzymes such as superoxide dismutase (SOD), peroxidase (POD), catalase (CAT), and ascorbate peroxidase (APX), as well as metallothioneins and phytochelatins [17,28,29,30,31,32].

In comparison with AMF, the use of yeasts as plant growth-promoting (PGP) inoculants has not been extensively investigated [33,34]. Despite the above, a wide range of yeasts exhibit PGP characteristics, including phytohormone synthesis, phosphate solubilization, siderophore production, exopolysaccharide production, and the enzymatic activity of 1-aminocyclopropane-1-carboxylic acid (ACC) deaminase [34,35,36,37,38]. Moreover, several yeast strains have been described as tolerant to toxic elements, such as Al, Cd, Cu, Hg, Pb, Zn, As, Sb, and Si [37,38,39]. Certain yeasts have also shown positive effects on the formation of AMF structures in both roots and soil [40,41]. Although the above studies suggest the potential usefulness of these organisms in phytoremediation strategies, a deep understanding of the effects induced by the joint application of AMF and yeast in mine tailings is necessary. 

Here, we hypothesize that the combined effect of both biotechnological tools (AMF and yeast inoculation) in the presence of compost can improve phytoremediation processes in mine tailings by increasing plant growth, reducing Cu bioavailability, and modulating the antioxidant response in the host plant. Based on the above, we aimed to determine (1) the effects of the combination of AMF and yeast inoculation on the growth of *O. picensis*, and (2) their effects on the antioxidant response when plants are grown in Cu mine tailings. Positive results can strongly support the sustainable management of Cu mine tailings by successfully establishing plant cover during the ongoing phytoremediation process.

## 2. Results

### 2.1. Characterization of PGP Traits in Cu-Tolerant Yeast

A total of ten yeast strains were tolerant to Cu concentrations up to 600 mg kg^−^^1^ and showed different PGP traits (Table 1). The highest phosphate solubilization index was observed for RC (2.86), followed by F3 (2.65) and E4 (2.57), while the lowest was observed for AB (2.22). The solubilization efficiency was assessed based on the scale formulated by Silva-Filho and Vidor [42], where values under 1.0 were classified as very low solubilizers, values from 1.0 to 2.0 were classified as low solubilizers, values from 2.0 to 3.0 were classified as medium solubilizers, and values higher than 3.0 were classified as high solubilizers. All yeast strains corresponded to the category of medium phosphate solubilizers. A quantitative analysis of P solubilization showed that E4 had the highest solubilizing capacity (0.68 mg mL^−^^1^), followed by AA (0.55 mg mL^−^^1^) and RC (0.48 mg mL^−^^1^). All isolated yeast strains displayed the capacity to produce EPS (Table 1). The highest EPS-producing strain was OM (1.34 g L^−^^1^), followed by AB (1.03 g L^−^^1^) and RG (0.89 g L^−^^1^). Among the isolates, the highest level of IAA production was observed for F4, with a final IAA concentration of 18.74 μg mL^−^^1^. The categorization of in vitro production showed three principal groups: lower producers with 1 to 10 µg/mL IAA, medium producers with levels of 11 to 20 µg/mL IAA, and higher producers with levels of 21 to 30 µg/mL IAA [43]. The isolates of strains F4, PM, and OM were in the category of medium producers, while the remaining isolates corresponded to low producers. For siderophore production, only strains E4, RD, and PM did not show discoloration on CAS plates. For ACC deaminase activity, all strains were able to grow with 3 mM of ACC as the unique nitrogen source. Based on the PGP characteristics, we selected strains F4 and OM, as they produced the highest levels of IAA and EPS, respectively. Additionally, both were capable of phosphate solubilization and exhibited siderophore production and ACC deaminase activity. Subsequently, strains F4 and OM were sequenced and identified as *Rhodotorula mucilaginosa* (RPMT02; Genbank accession number OQ821986) and *Meyerozyma guilliermondii* (RPMT01; Genbank accession number OQ821980), respectively.

### 2.2. AMF Determinations

In general, yeast inoculation did not affect AMF propagules or root colonization. In detail, neither MG nor RM inoculation influenced AMF colonization by *Claroideoglomus claroideum* (CC) in *O. picensis* roots. However, all plants showed around 30% AMF root colonization (Figure 1A), with MIX being the treatment with the highest AMF colonization (33.2%). Regarding T-GRSP production, values were found in the range of 0.20 to 0.23 mg g^−1^, with RM and MIX being the treatments with the highest values (Figure 1B). MG had the highest spore density, with a mean of 52 spores per 10 g of dry substrate (Figure 1C), and RM presented the highest hyphal density, with a mean of 18.6 cm g^−1^ (Figure 1D).

### 2.3. Plant Biomass Production and Photosynthetic Pigments

*O. picensis* plants presented classic features of Cu toxicity, such as chlorosis and necrosis, in the NM-NY treatment. Nevertheless, some plants were able to produce flowers, even under these stress conditions. The dry weights of the shoots and roots were affected by the interaction of AMF and yeast inoculation (AMF × Yeast; Figure 1 and Appendix A). The plants inoculated with CC had significant increases in shoot dry weight in all treatments compared with NY-NM (Figure 2A). Inoculation with *C. claroideum* produced a significant increase in shoot biomass (14.67 g) in the absence of yeast inoculation (NY-CC), with an approximately 2.8-fold increase compared with NY-NM. Yeast inoculation (MG, RM, and MIX) also caused significant increases in shoot dry weight in all treatments compared to NY-NM. Noticeably, MG-NM presented the highest shoot biomass production (17.94 g), with an increase of approximately 3.4-fold compared with NY-NM (Figure 2A). In addition, MG-NM treatment caused significant growth in the absence of AMF, with an increase after inoculation of approximately 1.5-fold compared with inoculation with MG-CC (Figure 2A). Regarding root biomass, all treatments caused significant increases compared with NY-NM (1.3 g). The highest root production was shown by MG-CC (6.75 g), with an increase of approximately 5-fold (Figure 2B) compared with NY-NM. MG-CC treatment also showed significant differences from the RM-NM (4.22 g) and MIX-CC (3.89 g) treatments.

Photosynthetic pigments (Chl a, Chl b, and carotenoids) were significantly affected by the interaction of AMF and yeast inoculation (AMF × Yeast; Figure 2 and Appendix A). The photosynthetic pigments showed significant increases in all treatments compared with NY-NM (Figure 3). For all pigments, the most noticeable increases in photosynthetic pigments were found in the MIX-CC treatment (1.79, 0.76, and 2.64 mg cm^−^^2^ for Chl a, Chl b, and carotenoids, respectively), with an increase of about 2.5-fold compared with the NY-NM treatment. 

### 2.4. Cu Concentrations in Tailings and Plant Tissues

In general, the total Cu concentration found in the tailings at the end of the assay was not affected by the interaction of AMF and yeast inoculation (AMF × Yeast; Figure 4A and Appendix A). However, a significant increase was observed in all yeast-inoculated treatments, except for in the RM-NM treatment, when compared to the control treatment NY-NM (Figure 4A). Inoculation with *M. guilliermondii* produced higher values of total Cu, with 193.3 and 202.96 mg kg^−^^1^ for MG-CC and MG-NM, respectively, compared to 170.54 mg kg^−^^1^ for NY-NM, representing increases in Cu retention of 13.7 and 19.38% (MG-CC and MG-NM, respectively) by the substrate compared to the absolute control. AMF × Yeast interactions were not significant. For DTPA-extractable Cu, the interaction of AMF and yeast inoculation was significant (AMF × Yeast; Figure 4B and Appendix A). Significant decreases were observed in all inoculated treatments compared to NY-NM (48.76 mg kg^−^^1^). Inoculation with either *C. claroideum*, *M. guilliermondii*, or *R. mucilaginosa* significantly reduced DTPA-extractable Cu in the mine tailings, especially when they were applied in combination. The lowest values of DTPA-extractable Cu were found in MIX-CC and RM-CC (21.31 and 21.28 mg kg^−^^1^, respectively), with a reduction of about 56% compared with NY-NM. 

In both the plant shoots and roots, the Cu concentrations were not significantly affected by the interaction of AMF and yeast (AMF × Yeast; Figure 5A,B and Appendix A). Regarding the plant shoots (Figure 5A), the Cu concentrations were significantly higher in plants inoculated with both yeasts (MIX-NM and MIX-CC), reaching values 30.93 and 36.32% higher than in the NY-NM treatment. In the roots (Figure 5B), the Cu concentrations were significantly higher in the NY-CC, MG-CC, and RM-CC treatments (204.01, 222.53, and 223.61 mg kg^−^^1^, respectively) compared to in the NY-NM treatment (102.26 mg kg^−^^1^), which means increases in Cu concentrations in the roots of 98.9, 116.96, and 118.02%, respectively. Inoculation with *C. claroideum* increased the Cu concentrations in the roots, except for in the MIX-CC treatment (155.16 mg kg^−^^1^), which had a lower concentration than the MIX-NM treatment (191.47 mg kg^−^^1^). For the shoot/root Cu translocation ratio, the interaction of AMF and yeast was significant (Figure 5C and Appendix A). The translocation ratio was higher in NY-NM with a value of 0.061. Significant reductions were observed in the NY-CC, MG-CC, and RM-CC treatments (0.032, 0.034, and 0.031, respectively) compared to the NY-NM treatment, resulting in a translocation reduction of about 45%. Inoculation with *C. claroideum* reduced the Cu translocation factor, except for in MIX-CC (0.054), which had a higher value than MIX-NM (0.046). 

### 2.5. Antioxidant Activities and Phenols

Regarding the enzymatic antioxidant activities, the interaction of AMF and yeast was significant only for SOD and APX (Figure 6 and Appendix A). The SOD activity was higher in NY-NM (56.72 EU mg protein^−^^1^) and was significantly affected by yeast inoculation in MG-NM and RM-NM (50.8 and 50.1 EU mg protein^−^^1^, respectively), which reduced the activity by approximately 10%. For CAT activity, the RM-NM and MIX-CC treatments showed significant differences from the NY-NM treatment (Figure 6B), with 48% and 68% higher CAT activity, respectively. The RM-NM and MIX-CC treatments also showed a significant increase in CAT with respect to the NY-CC treatment. APX activity showed significant differences only in MG-CC and MIX-CC compared with NY-NM. The co-inoculation of MG with CC showed an increase of 139%, while MIX-CC showed a 108% increase in APX activity compared with NY-NM.

For the total phenols in the shoots, no significant differences were observed for the interaction of AMF and yeast (Figure 7A and Appendix A), and only the RM-CC treatment (GAE 387.3 µg g^−^^1^) showed a significant difference from the NY-NM treatment (GAE 591.78 µg g^−^^1^), with a reduction of 34.5%. For non-enzymatic antioxidant activity in the shoots, the interaction of AMF and yeast was significant for DPPH and ABTS activities (Figure 7B,C and Appendix A). For DPPH activity in the shoots (Figure 7B), NY-NM showed higher activity, reaching TE 15.53 µmol g^−^^1^. All treatments showed significant decreases in DPPH activity compared to the NY-NM treatment, except for the NY-CC and MG-CC treatments. The most significant reductions were observed in the RM-CC, MIX-NM, and MIX-CC treatments, with reductions of 44.7%, 43.7%, and 41.3%, respectively. In addition, the previously mentioned treatments had significant reductions compared to yeast inoculation in the absence of CC (RM-NM and MG-NM). For CUPRAC activity in the shoots (Figure 7C), the NY-NM (TE 54.01 µmol g^−^^1^) and MIX-CC (TE 53.08 µmol g^−^^1^) treatments showed the highest activities. A significant reduction was observed for RM-CC with a decrease of 40.89% compared to NY-NM. For ABTS activity in the shoots (Figure 7D), only RM-CC (TE 7.23 µmol g^−^^1^) showed a significant difference from NY-NM (TE 8.65 µmol g^−^^1^), with a reduction of 16%.

Regarding the total phenols in the roots, the interaction of AMF and yeast was significant (Figure 8A and Appendix A). The NY-NM treatment showed the highest phenol content (GAE 684.96 µg g^−^^1^) and significant reductions were shown in the RM-NM, MG-NM, NY-CC, RM-CC, and MIX-NM treatments (58.13%, 5.01%, 54.90%, 52.62%, and 31.17%, respectively). Only MIX-CC and MG-CC showed phenol contents similar to NY-NM. For non-enzymatic antioxidant activity in the roots, the interaction of AMF and yeast was significant for DPPH activity and ABTS activity (Figure 8B and Appendix A). Only the RM-NM treatment (TE 4.90 µmol g^−^^1^) had a significant reduction compared with the NY-NM treatment (TE 9.19 µmol g^−^^1^), with a reduction of 46.6%. For CUPRAC activity in the roots (Figure 8C), the MG-NM (TE 25.93 µmol g^−^^1^), RM-NM (TE 27.89 µmol g^−^^1^), and RM-CC treatments (TE 41.70 µmol g^−^^1^) had significant reductions of 74.75%, 72.83% and 59.39%, respectively, compared with the NY-NM treatment (TE 102.70 µmol g^−^^1^). For ABTS activity in the roots (Figure 8D), only RM-NM (TE 5.77 µmol g^−^^1^) and MG-NM (TE 6.34 µmol g^−^^1^) showed significant differences from NY-NM (TE 10.64 µmol g^−^^1^), with reductions of 45.71% and 40.42%, respectively. 

### 2.6. Multivariate Associations

The PCA showed the formation of highly homogeneous groups of experimental variables and treatments (Figure 9). PC1 and PC2 explained 36.6% and 16.8% of the total variance, respectively. The confidence ellipsoids showed the separation of eight well-defined groups of inoculation state. In general, the fungi inoculation treatments (MG, RM, MIX, and CC) clustered together on negative PC1 and were related to better plant performance traits such as higher dry weights of plant shoots and roots, increased production of photosynthetic pigments, total Cu in mine tailings, total Cu in plants (especially in roots), and enzymatic activity of CAT and APX. The non-inoculated CC treatment mainly appeared on negative PC2, except for MIX-NM, which was on positive PC2. The PCA showed a clear separation of the NM-NY treatment with respect to all other treatments, and it was highly related to PC1, which was associated with a high concentration of DTPA-extractable Cu, the Cu translocation factor of shoots/roots, SOD activity, total phenols, and non-enzymatic antioxidant activity in shoots and roots (CUPRAC, DPPH, and ABTS).

## 3. Discussion

The total Cu concentration in the mine tailings was near to 400 mg kg^−^^1^, which is within the range of other mining sites described in the north and central areas of Chile [17,44]. However, in Chile, several Cu-polluted areas present higher concentrations, even in a range from 2000 to 4300 mg Cu kg^−^^1^ [44,45]. Some endemic metallophytes present in these areas can grow under toxic conditions, developing different mechanisms to tolerate heavy metal(loid) levels in the soil [17]. In this sense, *O. picensis* has been described as a Cu-tolerant metallophyte, emerging as an alternative for the phytoremediation of Cu-polluted environments [16,17,24,25]. Moreover, we confirm herein that *O. picensis* can successfully grow in Cu mine tailings conditioned with medium levels of organic materials [16], regardless of the use of other biotechnologies, such as inoculation with AMF or other PGP yeasts. However, the results obtained here demonstrate that inoculation with either *C. claroideum*, *M. guilliermondii*, *R. mucilaginosa*, or a combination of these fungi caused significant increases in the growth of *O. picensis*, from 2- to 3-fold in shoots and 3- to 5-fold in roots. 

The beneficial effects of AMF on phytoremediation processes have been previously described, highlighting an increase in plant growth, the promotion of better nutritional status, and a decrease in heavy metal concentrations in the overall plant [18,24,46,47]. Notably, inoculation with the yeasts *M. guilliermondii* and *R. mucilaginosa*, either alone or in combination, produced an effect on plant growth similar to that produced by inoculation with *C. claroideum*. This demonstrates the yeasts’ capacity to act as efficient plant-promoting microorganisms in the presence of toxic elements such as Cu.

Both yeasts produced IAA, which is well known for primarily stimulating the proliferation of lateral roots in plants and increasing the root surface, as well as the plant’s capability to absorb most water and soil mineral nutrients [48], which is especially desirable in substrates such as tailings, recognized as very nutrient-poor environments. Another PGP trait displayed by the yeasts was ACC deaminase production, which can diminish global stress effects by reducing the levels of ethylene in plants. Besides acting as a growth regulator, ethylene can inhibit root elongation and act as a plant stress hormone. The yeast enzymatic activity of ACC deaminase can regulate the production of plant ethylene by metabolizing ACC into α-ketobutyric acid and ammonia, reducing its adverse effects on plants [49]. Although previous reports have suggested that yeast inoculation can have positive effects on the formation of AMF structures in roots and soil [40,41], here, inoculation with yeasts did not show any effect on AMF root colonization capability, GRSP production, spore formation, or hyphal density (Figure 1).

The toxicity of heavy metals can lead to an increase in their concentrations in plant tissues. Although the use of AMF has been associated with plant tolerance, it is not necessarily linked to a reduction in the metal content of plants [50]. In some cases, tolerance to heavy metals is not due to a reduction in the uptake of these metals but rather to an increase in overall nutrition and general plant growth induced by AMF [51,52,53]. Despite this, we observed that AMF increased the total Cu concentration in the roots by up to 2 times. Additionally, *O. picensis* acted as a metal excluder, with TF values ranging from 0.061 in the non-inoculated treatment to 0.031 in the treatments with AMF. However, we observed that AMF inoculation increased the total Cu concentration in the root by up to 2 times. This could suggest a phytostabilizing effect by *C. claroideum*, which aligns with previous studies conducted with this AMF [16,18,54,55]. Regarding the Cu concentration in shoot tissues, we observed that, despite co-inoculation with *M. guilliermondii* and *R. mucilaginosa* showing an increase in Cu in shoots from 6.26 to 8.54 mg kg^−^^1^ compared with NY-NM (Figure 4), the obtained values were within the normal range of copper concentrations in plants [56].

However, the combined effect of AMF and yeast had a significant impact on the DTPA-extractable Cu in the mine tailings, resulting in a reduction of up to 56% compared with the non-inoculated treatment. Various effects at the soil–root interface have been reported for AMF inoculation [17,25,57,58]. These effects are primarily attributed to the presence of high-affinity proteins, including chitin, melanin, and glomalin, in the fungi. These proteins enable the accumulation of potentially toxic metal elements in the hyphae and even within spores, acting as biological barriers [18,26,57]. On the yeast front, the presence of exopolysaccharides (EPS) may influence the soil–root interface, thereby reducing the availability of Cu. The intricate role of microbial EPS production in enhancing soil aggregation, particularly under challenging conditions such as salinity and drought, has been well documented [59,60,61]. Recent studies have further underscored the significant biotechnological potential of EPS, highlighting its efficacy in both the sequestration and remediation of heavy metals [50,62,63].

Cu toxicity affects various biochemical and physiological processes in plants, due to several interactions expressed at the cellular and molecular levels [64,65,66]. In detail, a decrease in the chlorophyll concentration and membrane damage caused by oxidative stress are the main responses inducing toxicity and the loss of plant growth [67]. Here, we demonstrated that inoculation with either AMF, yeasts, or a combination of the two caused significant increases in photosynthetic pigments, especially when used in combination, increasing chlorophyll a, chlorophyll b, and carotenoid concentrations by about 2.5-fold compared with non-inoculated plants. The increase in plant growth concomitantly with the increase in photosynthetic pigments strongly suggests a reduction in the damage caused by reactive oxygen species (ROS), which are extensively produced in response to excess Cu [65,66,67,68].

One effect of chlorophyll recovery is an increase in the amount of light captured by leaves, which, in turn, reduces the possibility of membrane damage by ROS [68,69]. In this sense, SOD acts as the first line of defense to cope with ROS production, catalyzing the conversion of superoxide radical (O_2_^●–^) or singlet oxygen (1O_2_) to the harmless molecules hydrogen peroxide (H_2_O_2_) and molecular oxygen (O_2_) [70]. Here, for *O. picensis* plants, SOD showed high activity among the treatments. Only yeast inoculation with *M. guilliermondii* and *R. mucilaginosa* alone showed a reduction in SOD activity of about 10%. Additionally, H_2_O_2_ is still a potentially dangerous subproduct of oxygen metabolism because it is highly reactive with molecules containing Fe^2+^ or other transition metals through Fenton reactions, which results in the homolysis of H_2_O_2_ to two harmful -OH radicals [71]. However, H_2_O_2_ can be removed through the enzymatic actions of CAT and peroxidases [72]. Here, CAT activity was significantly increased by inoculation with *R. mucilaginosa* alone and co-inoculation with *C. claroideum* and *M. guilliermondii*, with an increase in enzyme activity of about 2-fold. Similarly, APX activity catalyzes the transformation of H_2_O_2_ in H_2_O using ascorbate as a hydrogen donor and produces monodehydroascorbate. In plants, enzymatic APX is found in several cellular compartments, such as chloroplasts, the cytosol, mitochondria, peroxisomes, and microbodies [73]. In our study, the APX activity of *O. picensis* plants showed a significant increase when *M. guilliermondii* was co-inoculated with *C. claroideum*, with and without the presence of *R. mucilaginosa*. In general, plants co-inoculated with AMF and yeast showed significantly increased APX and CAT activities. This response was also directly correlated with photosynthetic pigments and negatively correlated with the availability of Cu in the mine tailings.

In terms of the non-enzymatic antioxidant response, it is well established that phenolic compounds possess antioxidant capacity, protecting plant cells from the detrimental effects of oxidative damage under various stress conditions [60,71,74]. When subjected to metal stress, plants typically enhance the biosynthesis of phenolic compounds as a defense mechanism against oxidative stress [75].

In the shoots of *O. picensis*, we observed a significant reduction in total phenols only with co-inoculation of *C. claroideum* and *R. mucilaginosa*, resulting in a corresponding decrease in CUPRAC and ABTS antioxidant activities in the shoot. Conversely, inoculation with AMF and PGP yeasts alone led to a noteworthy 50% reduction in total phenols in the roots when compared to non-inoculated plants. This reduction in total phenols in both the shoots and roots may serve as an alleviation signal produced by the fungi under Cu toxicity, as plants typically respond to metal stress by increasing the biosynthesis of phenolic compounds as a protective mechanism [75]. Despite this, the plants colonized by *C. claroideum* did not show significant changes in the non-enzymatic antioxidant response. In this sense, inoculation with PGP yeasts caused a significant reduction in the antioxidant activities of DPPH, ABTS, and especially CUPRAC in the roots, which presented a great reduction in activity among the treatments. This suggests that, in the presence of Cu toxicity, *O. picensis* produces a higher level of phenolic compounds, such as flavonols or hydroxycinnamic acid derivatives, which respond better to CUPRAC assays [76,77]. This coincides with the finding that indicates an increase in the content of flavonoids in relation to the abundance of metals [78]. In general, flavonols can improve metal chelation mechanisms, reducing the content of hydroxyl radicals in plant cells [79]. This evidence suggests that the addition of PGP yeasts could modify the production of these phenols in *O. picensis*, alleviating the oxidative effects exerted by Cu toxicity and providing a good option for inclusion as a bioinoculant to generate plant cover in Cu-contaminated environments such as Cu mine tailings.

Although the AMF × Yeast interaction was significant for shoot growth, photosynthetic pigments, and enzymatic and non-enzymatic antioxidant activities (Appendix A), no synergistic effect of fungal inoculation was observed. In contrast, *M. guilliermondii* seemed to exhibit better performance when inoculated alone rather than when co-inoculated with *R. mucilaginosa* and *C. claroideum*, suggesting the co-occurrence of rhizosphere functions that may influence certain competitive relationships [55]. This indicates that the use of consortia as a tool for rhizosphere management could be analyzed in depth while also searching for possible antagonistic relationships.

## 4. Materials and Methods

### 4.1. Mine Tailings, Compost, Biological Material, and Procedures for Physicochemical Characterization

The mine tailings used in this study were collected from the non-operative reservoir Piuquenes, located in the Aconcagua Valley, Los Andes, Valparaíso Region (32°59′47.96″ S; 70°15′14.16″ W). Additionally, we used compost obtained from La Pintana municipality (33°34′60″ S; 70°37′60″ W) through the Direction of Environmental Management (DIGA), and this was produced using organic waste from the commune. This compost had previously been used in copper mining tailings and was applied as a foundation for the development of *O. picensis*, as reported by Pérez et al. [16]. The physicochemical characteristics of the mine tailings and compost were previously determined [16]. Briefly, the mine tailings had an acidic pH of 4, a total organic matter content of 0.3%, and a high total Cu concentration of 396 mg kg^−^^1^. Additionally, the DIGA compost had an alkaline pH of 8.5, a total organic matter content of 27.8%, and contributed NPK to the mine tailings.

The seeds of mature *O. picensis* plants were obtained from the ecosystem surrounding the Ventanas smelter installation in the Puchuncaví Valley (Valparaíso Region, Central Chile). The *Claroideoglomus claroideum* (CC) inoculum was provided by Centro de Investigación en Micorrizas y Sustentabilidad Agroambiental (CIMYSA; Universidad de La Frontera. Temuco). The CC fungus was isolated from the rhizosphere of wheat plants growing in volcanic soil (La Araucanía Region, central-south Chile). This inoculum had been previously used with *O. picensis* in Cu-polluted soils and with *Lactuca sativa* under salinity stress [16,54,80]. 

The seeds were surface-sterilized with 2% chloramine-T solution for 5 min and rinsed thoroughly with distilled water. The sterilized *O. picensis* seeds were incubated with a yeast suspension containing 1 × 10^6^ viable cells mL^−^^1^ for 1 h [81]. The sterile distilled water was replaced for the control seeds. At sowing time, 10 g of CC inoculum, containing approximately 200 spores per gram, was added to the respective treatments. To produce non-AMF-colonized plants, we used the same inoculum after autoclave sterilization at 121 °C for 30 min. Before the greenhouse assay, the mine tailings were manually mixed with compost at 5% *w*/*w* and stored for two weeks. This compost concentration allowed for the improvement of the pH of mine tailings, reduced DTPA-extractable Cu, enhanced the growth of *O. picensis*, and improved the mycorrhizal parameters of CC [16]. After 15 days of seedling emergence, three plantlets were placed in sterile pots (1 L) filled with mine tailings. At this point, a second inoculation was carried out; 1 mL of fresh yeast suspension treatment containing 1 × 10^6^ viable cells mL^−^^1^ was used as inoculum for each plant. Non-inoculated treatments used the same quantity of sterile distilled water instead.

### 4.2. Isolation and Identification of Cu-Tolerant Yeast with PGP Activities

Yeast strains were isolated from Cu mine tailings using a solid YPD medium (yeast extract, peptone, dextrose, and agar at 10, 20, 20, and 35 g L^−^^1^, respectively, pH 4.8) supplemented with 25 µg mL^−^^1^ Rose Bengal and 200 µg mL^−^^1^ chloramphenicol. To isolate the yeasts, 2 g of each Cu mine tailing sample was transferred to a flask with 20 mL of sterile 0.9% *w*/*v* NaCl saline solution and shaken at 200 rpm for 30 min. The supernatant was subjected to serial dilution, and 100 µL of each sample was spread on solid YPD medium. Plates were incubated at 30 °C for 72 h, and single colonies were purified via streaking. The purified yeast strains were suspended in liquid YPD medium supplemented with 25% *v*/*v* glycerol and stored at −80 °C for further analysis.

To determine Cu tolerance, 100 µL of yeast isolate was inoculated on YPD agar amended with CuSO_4_ to reach final Cu concentrations of 0, 400, and 600 mg Cu L^−^^1^. Plates were incubated at 30 °C for 72 h, where visible colonies on the plates were considered Cu-tolerant yeast strains and analyzed for PGP traits. To determine phosphate solubilization activity, the yeast isolates were inoculated on Pikovskaya agar medium [82] and incubated at 30 °C for 7 days. To measure the P solubilization index (PSI), the colony diameter and halo zone were measured each day [83]. The capacity for P solubilization was quantified in Pikovskaya broth using the chlorostannous-reduced molybdophosphoric acid blue method [84]. In brief, the yeast isolates were inoculated in PKV broth and incubated at 30 °C for 7 days with shaking at 125 rpm. After incubation, 20 mL of each culture was centrifuged at 3000× *g* for 15 min. To determine soluble P in the supernatant, 10 mL of chloromolybdic acid was added to 10 mL of supernatant, along with 5 drops of chlorostannous acid, and then adjusted to a total volume of 50 mL with distilled water. The developed blue color was measured at 600 nm. The amount of solubilized P was calculated using a calibration curve of KH_2_PO_4_. Siderophore production was determined through the color change of the solid blue medium Chrome-azurol S (CAS). After solidifying, paper disks were placed on plates, and 40 µL of yeast isolate was dropped onto the disk. The inoculated plates were incubated at 30° for 7 days. The development of a yellow to orange halo around the yeast colonies indicated siderophore production [85]. To determine ACC deaminase activity, the yeast isolates were inoculated on solid Dworkin and Foster salt medium and spread with 3 mM of ACC as the unique nitrogen source [86]. The inoculated plates were incubated at 30 °C for 7 days. Yeast growing on the selective medium confirmed the capability to utilize ACC as a nitrogen source. The inoculated plates were incubated at 30 °C for 7 days [87]. To quantify IAA, the yeast isolates were inoculated on YPD medium supplemented with 100 μg mL^−^^1^ of tryptophan and incubated at 30 °C for 72 h with shaking at 125 rpm. After incubation, 5 mL of each culture was centrifuged at 3000× *g* for 15 min. Then, 2 mL of Salkowsky’s reagent (2% of 0.5 M FeCl_3_ in 35% of perchloric acid) was added to 2 mL of supernatant and incubated at 30 °C in the dark for 1 h. The indole-3-acetic acid (IAA) concentrations of each culture (color red) were determined via spectrophotometry at 540 nm using a standard curve of IAA [88,89]. To determine exopolysaccharide (EPS) production [36], yeast isolates were grown on EPS production medium (per liter, sucrose 50 g, peptone 0.6 g, yeast extract 0.4 g, K_2_HPO_3_ 5 g, and MgSO_4_ · 7H_2_O 0.4 g, pH 6.8) at 30 °C for 7 days. Then, the cells were separated from the broth culture via centrifugation at 4000 rpm for 30 min at 4 °C. The resulting precipitated cells were washed with distilled water and dried at 80 °C to a constant weight to obtain the biomass. The EPS was precipitated from the cell-free supernatants by adding a 3-fold volume of cold acetone, followed by centrifugation and evaporation. The crude EPS was then dissolved in milli-Q water and freeze-dried, and its concentration was determined using the phenol–sulfuric acid method at 481 nm with glucose solution as the standard [90]. All PGP analyses were performed in triplicate, and for each analysis, fresh YPD yeast cultures with an adjusted OD of 0.6 were used. All spectrophotometric measurements were conducted using the EPOCH microplate spectrophotometer (BioTek Instruments, Inc., Winooski, VT, USA). Based on the above procedures and results, two yeast strains were selected as inoculants for further greenhouse experiments.

For molecular identification of the yeasts, total genomic DNA was extracted from 2 mL of 7-day-old samples of each yeast strain cultured on YPD broth medium, using the Wizard^®^ Genomic DNA Purification Kit (Promega, Madison, WI, USA) according to the manufacturer’s instructions. Polymerase chain reaction (PCR) was performed using GoTaq^®^ DNA Polymerase (Promega, Madison, WI, USA) at 25 μL per reaction with a final concentration of 1× colorless buffer; 3 mM MgC_l2_; 1 mM dNTPs; 0.4 μM of each primer, namely the forward primer ITS3 (5′-GCATCGATGAAGAACGCAGC-3′) and the reverse primer ITS4 (5′-TCCTCCGCTTATTG ATATGC-3′) corresponding to the internal transcriber spacer 2 (ITS2) region; 1.25 U per reaction of GoTaq^®^ DNA Polymerase; and ~0.25 μg of genomic DNA. The amplification conditions were 94 °C for 5 min for initial DNA denaturation, 35 cycles at 94 °C for 30 s, 60 °C for 30 s, and 72 °C for 30 s, and a final elongation step at 72 °C for 5 min. The amplified products were analyzed via gel electrophoresis. The PCR products were sequenced from both directions using the forward and reverse primers on an ABI PRISM 3500xL automated DNA sequencer (Applied Biosystems, Foster City, CA, USA), using the sequencing service of Pontificia Universidad Católica, Santiago, Chile. The sequences were assembled and manually edited in BioEdit v.7.2.5. The resulting nucleotide sequences were analyzed using the BLAST program [91], and, subsequently, they were compared and submitted to the NCBI GenBank database.

### 4.3. Growth Conditions 

The *O. picensis* plants were grown under greenhouse conditions using a 16:8 light/dark photocycle, 18:26 °C night/day temperatures, and 50% relative humidity. The germinated seeds of *O. picensis* were maintained under the same conditions in trays for three weeks after their emergence. The experiment was maintained under the same conditions for 60 days, with the pots being randomized weekly to avoid block effects. All pots were water-adjusted every two days with 100 mL of distilled water.

### 4.4. Cu Content in Mine Tailings and Plant Tissues

The total Cu concentration in the mine tailings was determined via digestion of the samples in boiling nitric acid, followed by the addition of perchloric acid [92], and analyzed using an atomic absorption spectrophotometer (AAS, Unicam SOLAAR, mod. 969). For the determination of available Cu in the mine tailings, a solution of DTPA-CaCl_2_-TEA at pH 7.3 was used as the extractant [93] and available Cu was quantified using AAS, as described above. For the total Cu in plants, fresh shoots and roots were separated, dried at 65 °C for 48 h, and weighed. Subsequently, the plant tissues were ground, converted into ash using a muffle furnace at 550 °C, and digested using a H_2_O:HCl:HNO_3_ mixture of 8:1:1, *v*:*v*:*v* [54]. The plant extracts were used for total Cu determination using AAS, as described above. 

### 4.5. Plant Photosynthetic Pigments and Antioxidant Response

To determine the content of photosynthetic pigments, three leaf discs of 4 mm in diameter were extracted with a solution of 80% acetone at 4 °C for 24 h. The absorbance of the extracts was measured using an EPOCH microplate spectrophotometer (BioTek Instruments, Inc., Winooski, VT, USA) at wavelengths of 664, 647, and 430 nm for chlorophyll a (Chl a), chlorophyll b (Chl b), and carotenoids, respectively [94]. The concentrations of the pigments were calculated according to the equations reported by Lichtenthaler [95]. To obtain extracts for the enzymatic assays, 1 g of leaf sample was frozen in liquid nitrogen and subsequently ground in 3 mL of a solution consisting of 0.1 M phosphate buffer (pH 7.0) and 2.5% (*w*/*v*) polyvinylpyrrolidone (PVPP). The samples were centrifuged at 13,000× *g* for 15 min at 4 °C, and the supernatant was collected for subsequent enzymatic assays. The protein content of the enzyme extract was determined using the Bradford method [96]. The superoxide dismutase (SOD) activity assay was performed following the method of Beyer and Fridovich [97]. This assay measures the ability of SOD to inhibit the reduction of nitroblue tetrazolium (NBT) via the superoxide radicals generated photochemically, with the absorbance measured at 540 nm. SOD activity is expressed as enzyme unit (EU) per mg of protein, with one unit defined as the amount of protein causing a 50% decrease in SOD-inhibitable NBT reduction. The catalase (CAT) activity assay was performed according to the method described by Aebi [98]. The reaction mixture contained 20 mM H_2_O_2_ in 0.1 M phosphate buffer (pH 7.0). CAT activity was determined by measuring the decrease in absorbance at 240 nm for 3 min following the decomposition of H_2_O_2_ at 25 °C. The ascorbate peroxidase (APX) activity assay was performed according to the method described by Nakano and Asada [99], following the oxidation of ascorbate to dehydroascorbate at 290 nm at 25 °C. The reaction mixture contained 20 mM H_2_O_2_, 0.1 M phosphate buffer (pH 7.0), and 20 mM ascorbic acid.

For the quantification of phenolic compounds, 0.5 g of fresh shoots or root tissues was pulverized with liquid nitrogen, mixed with 1.5 mL of a solution of methanol/formic acid (97:3; *v*:*v*), and subjected to ultrasonic treatment for 60 s, followed by orbital agitation for 15 min at room temperature. Then, centrifugation at 4000× *g* for 10 min was performed to obtain the extract. All of the extracts were dried in a rotary evaporator and re-suspended in 1 mL of mobile phase (water:acetonitrile:formic acid; 92:3:5; *v*:*v*:*v*). The total phenol concentrations were assessed using the methodology outlined by Alonso et al. [100], adapted for a microplate reader as detailed in the study by Parada et al. [101]. Absorbance readings were taken at 750 nm using the EPOCH microplate spectrophotometer (BioTek Instruments, Inc., Winooski, VT, USA), with gallic acid employed as the reference standard. The results are expressed as milligrams of gallic acid per gram of fresh weight. Antioxidant activity was determined via the Cupric Ion Reducing Antioxidant Capacity (CUPRAC), 2,2′-azino-bis(3-ethylbenzothiazoline-6-sulfonic acid (ABTS), and 2,2-diphenyl-1-picrylhydrazyl (DPPH) methodologies, according to the descriptions by Parada [102]. The results are expressed as Trolox equivalents (TE). All antioxidant activities were measured using the EPOCH microplate spectrophotometer (BioTek Instruments, Inc., Winooski, VT, USA).

### 4.6. Arbuscular Mycorrhizal Fungal Measurements

Root colonization (%) by AMF was determined via visual observation using a stereomicroscope (40–60× magnification) of fresh root pieces (1 cm) obtained before the drying process. Fragments of about 1 cm were obtained at harvest, cleaned using KOH 10% *w*:*v*, and stained using trypan blue solution 0.05% *w*:*v* in lactic acid [102]. Colonization was calculated according to the gridline intersect method [103]. AMF spores were isolated from 10 g of substrate through wet sieving and the decanting method, followed by sucrose centrifugation at 2500 rpm for 10 min. After centrifugation, the supernatant was poured through 500 and 50 µm pore size meshes and quickly rinsed with tap water. The spores were transferred to a Petri dish and counted under a dissecting microscope [104]. The total GRSP (T-GRSP) was extracted from 1 g of substrate with 8 mL of 50 mM citrate buffer at pH 8.0 and autoclaved for 1 h at 121 °C. The supernatant was separated via centrifugation at 8000× *g* for 20 min and filtered through Whatman No. 1 paper. This process was carried out multiple times on the same sample, until the characteristic reddish-brown color of GRSP was no longer visible in the supernatant; then, all extracts from a soil sample were combined. The GRSP content in the crude extract was determined using the Bradford assay (BioRad Protein Assay; Bio Rad Laboratories, Des Plaines, IL, USA), with bovine serum albumin as the standard.

### 4.7. Experimental Design

An in vivo pot experiment with mine tailings was performed using a full-factorial randomized experimental design that included four levels of yeast inoculation: (1) non-yeast-inoculated plants (NY); (2) plants inoculated with *M. guilliermondii* (MG); (3) plants inoculated with *R. mucilaginosa* (RM); and (4) plants inoculated with a combination of both yeasts (MIX). In addition, two levels of inoculation with *C. claroideum* were used: (1) non-mycorrhizal-inoculated plants (NM) and (2) plants inoculated with *C. claroideum* (CC). The combination of the factors produced eight treatments with five replicates (n = 5), with three plants in each pot considered as one experimental unit. The plants were harvested at 60 days of growth after transplanting. It must be clarified that, in this study, we used a homogeneous level of compost (5% *w*/*w*), according to the previous results reported by Pérez et al. [16].

### 4.8. Statistics 

The main effects of yeast and AMF inoculation and their interaction on the growth and antioxidant response of *O. picensis* were statistically analyzed by means of two-way ANOVA. Treatments with significant differences were analyzed using HSD Tukey as a post-hoc test to compare the means between treatments. In addition, the data were also subjected to principal component analysis (PCA) to evaluate the multivariate effects of the established treatments and the relationships between variables. For all procedures, we considered *p* < 0.05 as statistically significant. The software SPSS 22.0 (IBM) and R statistics 3.5.1 were used for analyses.

## 5. Conclusions

Our findings demonstrate that, although *O. picensis* was capable of establishing itself, inoculation with AMF, PGP yeasts, or their combination had a positive impact on its growth. This effect was demonstrated by increased shoot and root biomasses, accompanied by higher levels of photosynthetic pigments.

Furthermore, our study confirmed that the application of AMF, PGP yeasts, or a combination of these fungi reduced the availability of Cu and enhanced the retention of Cu in recalcitrant fractions within mine tailings. These fungi could induce specific effects on the growth of *O. picensis*. Consequently, our outcomes suggest that AMF inoculation plays a pivotal role in phytostabilization, leading to higher Cu concentrations in roots and reduced Cu translocation from roots to shoots.

Conversely, the inoculation with PGP yeasts primarily influenced the enzymatic antioxidant responses of *O. picensis*, reducing the SOD activity and increasing the CAT and APX activities in the shoots. Additionally, although both AMF and PGP yeasts decreased the total phenols in the roots, only PGP yeasts significantly affected the non-enzymatic antioxidant responses, especially in terms of CUPRAC activity. This suggest a potential alteration in phenol biosynthesis induced by PGP yeasts, possibly in the form of flavonols within *O. picensis* plants.

Notably, co-inoculation with these fungi did not result in an additive effect on the growth of *O. picensis* or on the status of AMF colonization. This observation raises questions regarding the underlying mechanisms that may lead to competitive relationships in the rhizosphere. Improving the understanding of the dynamics between these fungal species and their influence on plant responses could be crucial for the successful development of bioinoculants in Cu-contaminated environments, such as mine tailings, and it could provide valuable insight for the restoration of sustainable plant cover in challenging ecosystems.

## Figures and Tables

**Figure 1 plants-12-04012-f001:**
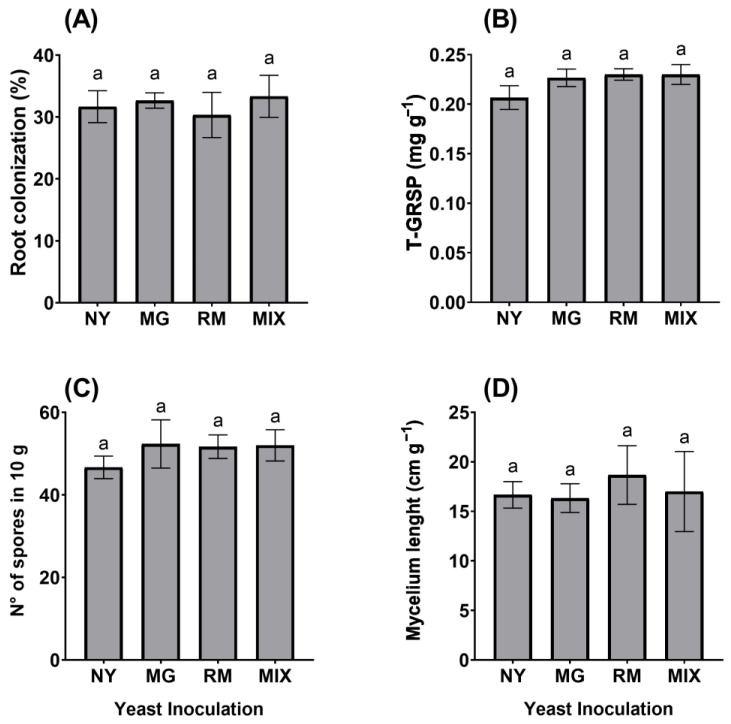
Arbuscular mycorrhizal fungal parameters of *Oenothera picensis* plants growing in Cu mine tailings and inoculated with the fungus *Claroideoglomus claroideum*, the yeasts *Meyerozyma guilliermondii* (MG), *Rhodotorula mucilaginosa* (RM), with both (MIX), or non-inoculated with yeast (NY). Percentage of root colonization (**A**), total glomalin-related soil protein (T-GRSP) (**B**), number of spores in 10 g of substrate (**C**), and mycelium length (**D**). Values are expressed as the mean ± standard error. Different letters indicate significant differences between means according to Tukey’s HSD test (*p* < 0.05, n = 5).

**Figure 2 plants-12-04012-f002:**
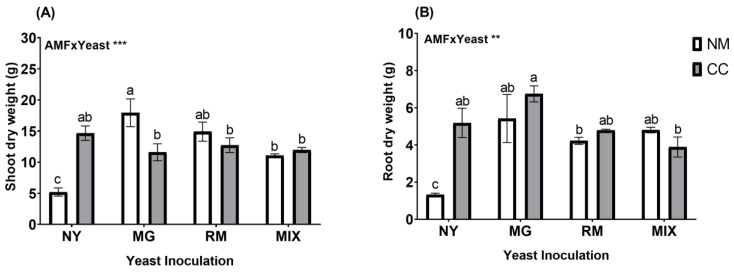
Dry weights of shoots (**A**) and roots (**B**) of *Oenothera picensis* plants growing in Cu mine tailings and inoculated with the fungus *Claroideoglomus claroideum* (CC) or not (NM) and the yeasts *Meyerozyma guilliermondii* (MG), *Rhodotorula mucilaginosa* (RM), or both (MIX). Values are expressed as the mean ± standard error. Different letters indicate significant differences between means according to Tukey’s HSD test (*p* < 0.05, n = 5). The significance of the source of interaction (AMF × Yeast) was determined through two-way ANOVA. When no significant interaction between factors was observed, a significant source was indicated (*p*-values: ** *p* ≤ 0.01; *** *p* ≤ 0.001).

**Figure 3 plants-12-04012-f003:**
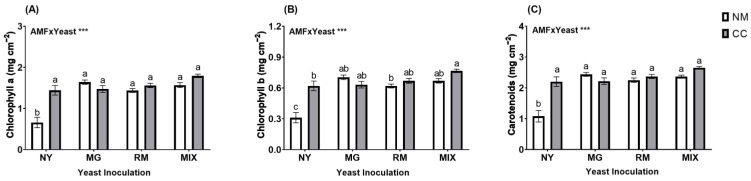
Chlorophyll a (**A**), chlorophyll b (**B**), and carotenoid (**C**) concentrations in leaves of *Oenothera picensis* plants growing in Cu mine tailings and inoculated with the fungus *Claroideoglomus claroideum* (CC) or not (NM) and the yeasts *Meyerozyma guilliermondii* (MG), *Rhodotorula mucilaginosa* (RM), or both (MIX). Values are expressed as the mean ± standard error. Different letters indicate significant differences between means according to Tukey’s HSD test (*p* < 0.05, n = 5). The significance of the source of interaction (AMF × Yeast) was determined through two-way ANOVA. When no significant interaction between factors was observed, a significant source was indicated (*p*-values: *** *p* ≤ 0.001).

**Figure 4 plants-12-04012-f004:**
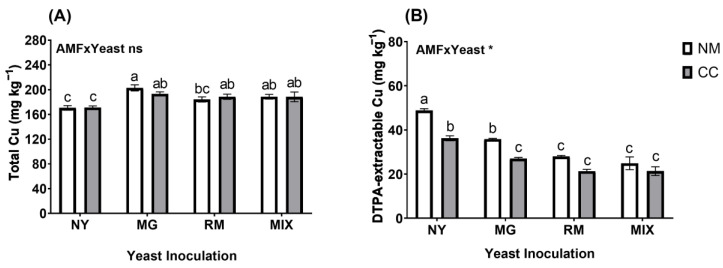
Total Cu concentrations in mine tailings (**A**) and DTPA-extractable Cu (**B**) after harvest of *Oenothera picensis* plants growing in Cu mine tailings and inoculated with the fungus *Claroideoglomus claroideum* (CC) or not (NM) and the yeasts *Meyerozyma guilliermondii* (MG), *Rhodotorula mucilaginosa* (RM), or both (MIX). Values are expressed as the mean ± standard error. Different letters indicate significant differences between means according to Tukey’s HSD test (*p* < 0.05, n = 5). The significance of the source of interaction (AMF × Yeast) was determined through two-way ANOVA. When no significant interaction between factors was observed, a significant source was indicated (*p*-values: ns, not significant; * *p* ≤ 0.05).

**Figure 5 plants-12-04012-f005:**
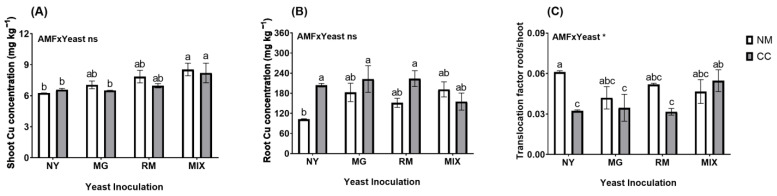
Total Cu concentrations in shoots (**A**), roots (**B**), and translocation factor shoot/root (**C**), of *Oenothera picensis* plants growing in Cu mine tailings and inoculated with the fungus *Claroideoglomus claroideum* (CC) or not (NM) and the yeasts *Meyerozyma guilliermondii* (MG), *Rhodotorula mucilaginosa* (RM), or both (MIX). Values are expressed as the mean ± standard error. Different letters indicate significant differences between means according to Tukey’s HSD test (*p* < 0.05, n = 5). The significance of the source of interaction (AMF × Yeast) was determined through two-way ANOVA. When no significant interaction between factors was observed, a significant source was indicated (*p*-values: ns, not significant; * *p* ≤ 0.05).

**Figure 6 plants-12-04012-f006:**
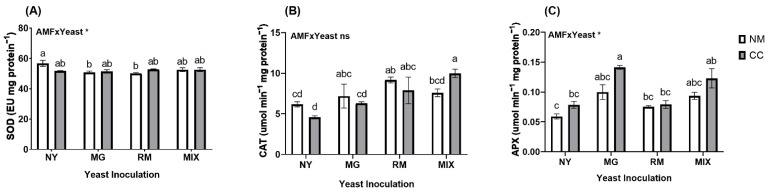
Antioxidant activity of (**A**) superoxide dismutase activity (SOD), (**B**) catalase (CAT), and (**C**) ascorbate peroxidase activity (APX) of *Oenothera picensis* plants growing in Cu mine tailings and inoculated with the fungus *Claroideoglomus claroideum* (CC) or not (NM) and the yeasts *Meyerozyma guilliermondii* (MG), *Rhodotorula mucilaginosa* (RM), or both (MIX). Values are expressed as the mean ± standard error. Different letters indicate significant differences between means according to Tukey’s HSD test (*p* < 0.05, n = 5). The significance of the source of interaction (AMF × Yeast) was determined through two-way ANOVA. When no significant interaction between factors was observed, a significant source was indicated (*p*-values: ns, not significant; * *p* ≤ 0.05).

**Figure 7 plants-12-04012-f007:**
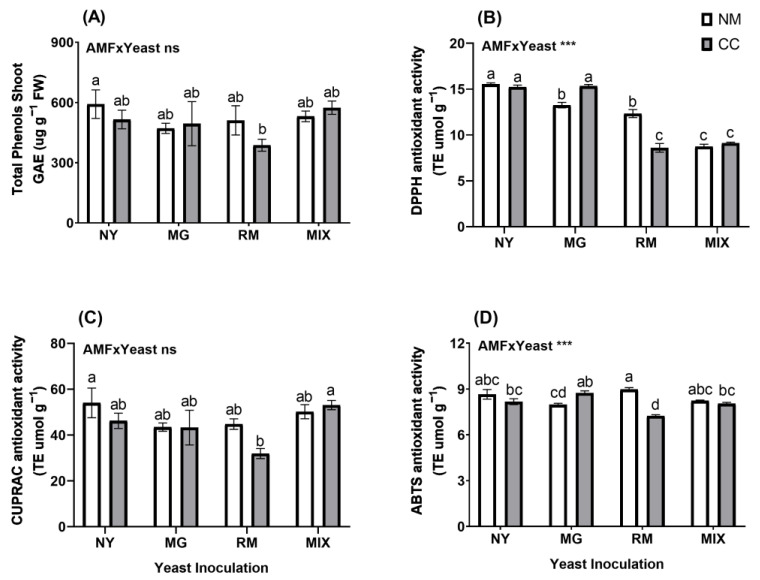
Total shoot phenolic compound concentrations (**A**) and antioxidant activity determined by DPPH (**B**), CUPRAC (**C**) and ABTS (**D**) of *Oenothera picensis* plants growing in Cu mine tailings and inoculated with the fungus *Claroideoglomus claroideum* (CC) or not (NM) and the yeasts *Meyerozyma guilliermondii* (MG), *Rhodotorula mucilaginosa* (RM), or both (MIX). Values are expressed as the mean ± standard error. Different letters indicate significant differences between means according to Tukey’s HSD test (*p* < 0.05, n = 5). The significance of the source of interaction (AMF × Yeast) was determined through two-way ANOVA. When no significant interaction between factors was observed, a significant source was indicated (*p*-values: ns, not significant; *** *p* ≤ 0.001).

**Figure 8 plants-12-04012-f008:**
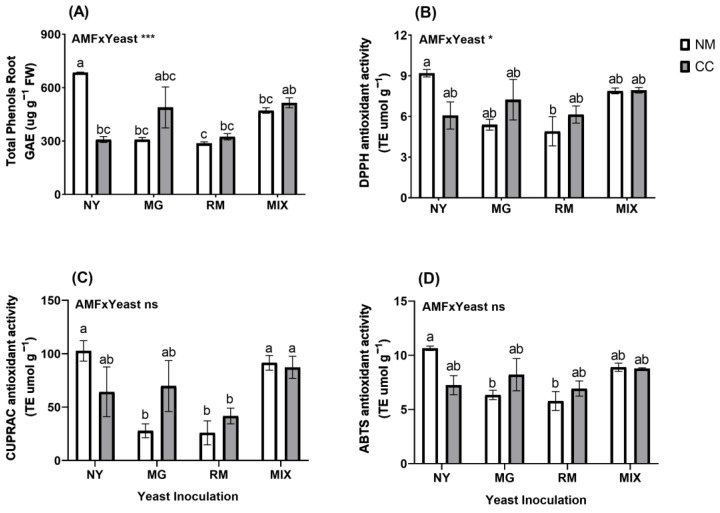
Total root phenolic compound concentrations (**A**) and antioxidant activity determined by DPPH (**B**), CUPRAC (**C**) and ABTS (**D**) of *Oenothera picensis* plants growing in Cu mine tailings and inoculated with the fungus *Claroideoglomus claroideum* (CC) or not (NM) and the yeasts *Meyerozyma guilliermondii* (MG), *Rhodotorula mucilaginosa* (RM), or both (MIX). Values are expressed as the mean ± standard error. Different letters indicate significant differences between means according to Tukey’s HSD test (*p* < 0.05, n = 5). The significance of the source of interaction (AMF × Yeast) was determined through two-way ANOVA. When no significant interaction between factors was observed, a significant source was indicated (*p*-values: ns, not significant; * *p* ≤ 0.05; *** *p* ≤ 0.001).

**Figure 9 plants-12-04012-f009:**
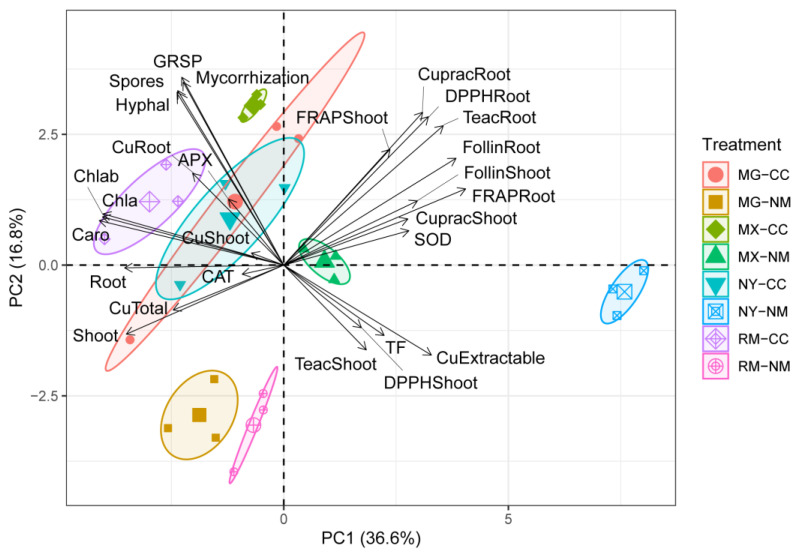
Principal component (PC) analysis for yeast-inoculated plants with AMF addition on plants growing in Cu mine tailings. The PCs are used for the ordination of the experimental variables and all of the individuals according to the treatments established, based on dry biomass production (Root and Shoot), total Cu concentration in shoots and roots (CuShoot, CuRoot), Cu translocation factor root to shoot (TF), total Cu and available Cu on the mine tailing at the end of the assay (CuTotal, CuExtractable), mycorrhizal parameters as percentage of mycorrhization (Mycorrhization), hyphal length (Hyphal), number of spores (Spores), glomalin-related soil protein (GRSP), photosynthetic pigments chlorophyll A (Chla), chlorophyll b (Chlb) and carotenoids (Caro), shoot enzymatic antioxidant activity superoxide dismutase (SOD), ascorbate peroxidase (APX) and catalase (CAT), total phenolic compounds of shoots (FolinShoot) and roots (FolinRoot), DPPH antioxidant activity of shoots (DPPHShoot) and roots (DPPHRoot), Cuprac antioxidant activity of shoots (CupracShoot) and roots (CupracRoot), and Teac antioxidant activity of shoots (TeacShoot) and roots (TeacRoot).

**Table 1 plants-12-04012-t001:** Plant growth-promoting traits observed in the isolates of Cu-tolerant yeasts obtained from Cu mining tailings. Values are expressed as the mean ± standard error of three independent samples.

Isolate	PSI	Phosphate Solubilization (mg mL^−1^)	EPS Production (g L^−1^)	IAA Production (μg mL^−1^)	Siderophore Production	ACC Deaminase Activity
AA	2.30 ± 0.01 ^cd^*	0.55 ± 0.05 ^ab^	0.82 ± 0.15 ^ab^	9.75 ± 0.07 ^c^	+	+
AB	2.22 ± 0.01 ^d^	0.28 ± 0.03 ^b^	1.03 ± 0.07 ^ab^	8.86 ± 0.14 ^d^	+	+
E4	2.57 ± 0.03 ^b^	0.69 ± 0.08 ^a^	0.46 ± 0.01 ^b^	9.24 ± 0.07 ^cd^	−	+
F3	2.65 ± 0.02 ^ab^	0.43 ± 0.03 ^ab^	0.85 ± 0.12 ^ab^	8.67 ± 0.03 ^d^	+	+
F4	2.36 ± 0.03 ^bcd^	0.37 ± 0.06 ^b^	0.66 ± 0.03 ^ab^	18.74 ± 0.07 ^a^	+	+
RC	2.86 ± 0.09 ^a^	0.48 ± 0.06 ^ab^	0.84 ± 0.14 ^ab^	8.83 ± 0.03 ^d^	+	+
RD	2.42 ± 0.01 ^bcd^	0.33 ± 0.05 ^b^	0.67 ± 0.22 ^ab^	10.75 ± 0.07 ^b^	−	+
RG	2.48 ± 0.01 ^bcd^	0.43 ± 0.02 ^ab^	0.89 ± 0.15 ^ab^	8.90 ± 0.08 ^d^	+	+
OM	2.51 ± 0.04 ^b^	0.27 ± 0.02 ^b^	1.34 ± 0.16 ^a^	11.01 ± 0.21 ^b^	+	+
PM	2.30 ± 0.01 ^cd^	0.28 ± 0.08 ^b^	0.64 ± 0.11 ^b^	11.27 ± 0.14 ^b^	−	+

* Different letters over the bars indicate significant differences between means at *p* < 0.05 (one-way ANOVA, Tukey’s HSD post-hoc test). PSI: phosphate solubilization index; EPS: exopolysaccharides; IAA: indole-3-acetic acid; ACC: 1-aminocyclopropane-1-carboxylic acid.

## Data Availability

The data presented in this study are available on request from the corresponding author.

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
