# Peer review of "Beneficial Interactive Effects Provided by an Arbuscular Mycorrhizal Fungi and Yeast on the Growth of Oenothera picensis Established on Cu Mine Tailings"

_plants, 2023, doi:10.3390/plants12234012_

Round 1
Reviewer 1 Report
Comments and Suggestions for Authors
It is necessary to order the presentation of materials and methods, placing them after the introduction to have a general and specific vision of each methodology before analyzing the results. There are many works in the bibliography that show variability in the bioaccumulation and translocation coefficients in AMF, even when dealing with the same strain but varying the place and conditions of its isolation and propagation. This could be delved into in depth and not be categorical when talking about exclusion. phytostabilization or phytoextraction.
Author Response
Reviewer 1:
- It is necessary to order the presentation of materials and methods. placing them after the introduction to have a general and specific vision of each methodology before analyzing the results.
A: Regarding the order of the presentation of Materials and Methods, we would like to note that the arrangement follows the format specified by the Plants journal.
- There are many works in the bibliography that show variability in the bioaccumulation and translocation coefficients in AMF. even when dealing with the same strain but varying the place and conditions of its isolation and propagation. This could be delved into in depth and not be categorical when talking about exclusion. phytostabilization or phytoextraction
A: Thank you for your insightful comment. We have carefully considered your feedback and made improvements to the discussion section. We added relevant citations addressing the variability in AMF behavior. Moreover, we have incorporated additional information highlighting the potential of this fungus as a stabilizer. We acknowledge the importance of avoiding categorical statements and have taken steps to present a more nuanced perspective (L406-427).
Reviewer 2 Report
Comments and Suggestions for Authors
The authors studied the effects of yeasts and AMF on the phytostabilization of Cu in tailings, and found some interesting results. Overall, this manuscript can be accepted after major revisions. My concerns are below.
General comments:
First, this is a bi-factorial design experiment. I suggest two-way ANOVA to determine if there are significant interactive effects between yeasts and AMF.
Second, the language can be improved.
Third, this study did not explore the effect of compost. I can’t understand why add 5% compost in all the treatments. The compost may decrease the toxicity of Cu.
Specific comments
L64: I recommend to cite a review:
Wang F. Occurrence of arbuscular mycorrhizal fungi in mining-impacted sites and their contribution to ecological restoration: Mechanisms and applications. Critical Reviews in Environmental Science and Technology, 2017, 47(20): 1901-1957.
L94: Ref 33 can be removed, because it focuses on Pseudomonas citronellolis, which does not belong to yeasts.
Fig. 1A: y-axis, correct “mycorrhization” as “root colonization”
L243: correct “on” as “in”
Fig. 4B: correct “Extractable Cu DTPA” as “DTPA-extractable Cu”
L408: Please check if Refs [13] and [20] study AMF. A previous study has confirmed that AM inoculation decreases Cu concentration in plants.
Wang et al. Inoculation with arbuscular mycorrhizal fungus Acaulospora mellea decreases Cu phytoextraction by maize from Cu-contaminated soil. Pedobiologia, 2007, 51(2): 99-109.
L437: correct “on shoot” as “in shoot”; similarly, in L439, correct as “in plants”. Please check the whole manuscript.
Conclusions: This part can be more concise.
L755: remove “Science of the Total Environment”
L781: italicize “Pseudomonas citronellolis”
L782: italicize “Helianthus annuus”; check all the references.
Comments on the Quality of English Language
The language can be improved.
Author Response
Reviewer 2:
General comments:
- this is a bi-factorial design experiment. I suggest two-way ANOVA to determine if there are significant interactive effects between yeasts and AMF.
A: We've addressed your suggestion by modifying our graphs to include the AMDxYeast interaction and adding a supplementary table with significance values. These adjustments significantly enhance our work, offering a more comprehensive view of yeast-AMF relationships. We've also expanded the discussion to thoroughly explore the significance and implications of this interaction. We trust these changes align with your expectations and contribute to the overall improvement of our work.
- the language can be improved.
A: Regarding to the language improvement, we would like to inform you that we subjected our manuscript to the scrutiny of a certified English language reviewer. We ensured to address and incorporate all suggested corrections to enhance the linguistic quality of our manuscript.
- this study did not explore the effect of compost. I can’t understand why add 5% compost in all the treatments. The compost may decrease the toxicity of Cu.
A: Regarding your third point on the absence of exploration of compost effects and the inclusion of 5% compost in all treatments, we would like to provide clarification. In a prior study (Perez et al., ref. 16), we utilized compost at diverse levels, and corroborated that a concentration of 5% is the alternative most advantageous. In the current research, we replicated this usage due to its positive impact on the establishment of O. picensis in Cu mine tailings. Additional information on compost application has been incorporated into the Materials and Methods section to address this aspect comprehensively. Please see L510-511 and L529-532.
Specifics comentes:
- L64: I recommend to cite a review: “Wang F. Occurrence of arbuscular mycorrhizal fungi in mining-impacted sites and their contribution to ecological restoration: Mechanisms and applications. Critical Reviews in Environmental Science and Technology. 2017. 47(20): 1901-1957.”
- L94: Ref 33 can be removed. because it focuses on Pseudomonas citronellolis. which does not belong to yeasts.
- 1A: y-axis. correct “mycorrhization” as “root colonization”
- L243: correct “on” as “in”
- 4B: correct “Extractable Cu DTPA” as “DTPA-extractable Cu”
- L408: Please check if Refs [13] and [20] study AMF. A previous study has confirmed that AM inoculation decreases Cu concentration in plants. “Wang et al. Inoculation with arbuscular mycorrhizal fungus Acaulospora mellea decreases Cu phytoextraction by maize from Cu-contaminated soil. Pedobiologia. 2007. 51(2): 99-109”
- L437: correct “on shoot” as “in shoot”; similarly. in L439. correct as “in plants”. Please check the whole manuscript.
- Conclusions: This part can be more concise.
- L755: remove “Science of the Total Environment”
- L781: italicize “Pseudomonas citronellolis”
- L782: italicize “Helianthus annuus”; check all the references.
A: Thank you for your detailed and constructive comments. We have carefully addressed each of your specific suggestions:
- L64: We have included the recommended review by Wang (2017) to provide a comprehensive overview of arbuscular mycorrhizal fungi in mining-impacted sites.
- L94: Ref 33 has been removed and replaced with a reference more relevant to yeasts and their PGP characteristics (L835).
- 1A: The y-axis label has been corrected to "root colonization" as suggested.
- L243: The preposition "on" has been corrected to "in" as recommended.
- 4B: The label "Extractable Cu DTPA" has been corrected to "DTPA-extractable Cu."
- L408: References have been checked, now both studies focus on arbuscular mycorrhizal fungi (AMF). Additionally, Wang et al. (2007) has been included to support the statement about AM inoculation decreasing Cu concentration in plants. (L388)
- L437 and L439: The corrections "in shoot" and "in plants" have been made throughout the manuscript.
- Conclusions: The conclusion section has been revised for conciseness (L710-734).
- L755: "Science of the Total Environment" has been removed.
- L781: "Pseudomonas citronellolis" and "Helianthus annuus" have been italicized in accordance with your suggestion. All other scientific names in the references have been thoroughly reviewed and adjusted.
Reviewer 3 Report
Comments and Suggestions for Authors
Brief Summary
The manuscript plants-2695358 entitled “Beneficial interactive effect provided by an arbuscular mycorrhizal fungi and yeast in the growth of Oenothera picensis established on Cu mine tailings” represents an interesting and novel investigation, describing preliminary results about the interaction between PGPY strains and AMF as a sustainable tool in phytoremediation of heavy metal contaminated mine tailing.
The work hypothesis and the aim are well described but the introduction and the material and method sections need some major revisions. Also, the general quality presentation of the manuscript needs some improvements. See specific comments below.
Specific comments
· Abstract: Even if it is well-detailed, the abstract is too long (272 words with a maximum of 200 words). I suggest shortening the part from line 28 to line 3, avoiding repetition and maintaining the key elements.
· Introduction: The introduction section needs some major improvement for better clarification, cohesion and fluidity between the different sections. In particular, I suggest to describe better the first paragraph related to environmental pollution linked to mining activity, mine tailings, and heavy metal mine tailing contamination. The whole section also needs improvement in the English language. See below the specific comments:
o L51: replace high metal(loid) with heavy metal(loid). Revise this word in the whole manuscript.
o L53: I suggest specifying the mine wastes.
o L55-57: This statement, as it is presented, appears not well-linked to the first part. Extend the paragraph by adding more information about the sources of heavy metal contamination and explain better the scarcity of organic matter and nutrients.
o L76: correct the typo “that”, removing the italics
· Results: In general, this section is well described. Please, you can find some specific comments to improve the section:
o L133: add the dot.
o L130: I suggest specifying the choice to use these two yeast strains. Which were the criteria used? Did you choose them focusing on particular PGP traits? If yes, which ones? Are there ordered criteria, depending also on the soil characteristics, that you follow? Specify it.
o L153: Specify the name of the mycorrhiza Claroideoglomus claroideum before inserting the abbreviation in brackets (CC)
o L194: Correct “percentage” with the case letter
· Material and methods: The section is well structured and the experimental design is well described. However, my main concern is related to the analysis of the PGP traits. As reported in the manuscript, you use only qualitative methods to detect phosphate solubilization, ACC deaminase and siderophore production. These methods may be insufficient and misleading unless it is correlated with quantitative methods. I suggest further investigation to repeat the characterization of PGP traits using quantitative analysis to have a more precise estimation of the results. Still, the following elements need to be revised:
o L528-533: specify if you use sterile pots and soil during the in planta experiments.
o L531: add more information (like isolation sources etc...) about the AMF inoculum that you use
o L535: specify the choice to do the inoculation after 15 days, providing a reference.
o L536: correct 1x10-6 to 1x106
o L548: replace “replicated” with a more specific term (like purified)
o L639: specify which kind of modification you did, compared to the protocol mentioned above
o L657-663: specify which kind of modification you did, compared to the protocol mentioned above
· Discussion: In general the discussion section is well organized. However, you should revise some points, as follows:
o L392: correct the measure unit expression (mg kg-1)
o L426-431: I suggest moving this part in the conclusion section as a future perspective, as a base for further investigation.
o L446: change the comma with the dot in decimal numbers
· Conclusions: The authors presented a summary of the study. I suggest giving more details on the future research directions. Please, find below a specific comment:
o L735: "M. guilliermondii" should be in italics
Other comments
All the sections need improvements for the English language.

Comments on the Quality of English LanguageThe English language has to be significantly improved in all the manuscript's sections.
Author Response
Reviewer 3:
- Abstract: Even if it is well-detailed. the abstract is too long (272 words with a maximum of 200 words). I suggest shortening the part from line 28 to line 3. avoiding repetition and maintaining the key elements.
A: Thank you for your valuable feedback on our abstract. We have successfully revised the abstract, shortening it from 272 to 227 words.
- Introduction: The introduction section needs some major improvement for better clarification. cohesion and fluidity between the different sections. In particular. I suggest to describe better the first paragraph related to environmental pollution linked to mining activity. mine tailings. and heavy metal mine tailing contamination. The whole section also needs improvement in the English language. See below the specific comments:
- L51: replace high metal(loid) with heavy metal(loid). Revise this word in the whole manuscript.
- L53: I suggest specifying the mine wastes.
- L55-57: This statement. as it is presented. appears not well-linked to the first part. Extend the paragraph by adding more information about the sources of heavy metal contamination and explain better the scarcity of organic matter and nutrients.
- L76: correct the typo “that”. removing the italics
A: We appreciate your thorough review of the introduction section, and we have carefully addressed each of your specific comments:
- L51: We have replaced "high metal(loid)" with "heavy metal(loid)" throughout the manuscript, as suggested.
- L53: The term "mine wastes" has been specified as “mine tailings” for clarity.
- L55-57: We extended the paragraph by providing additional information about the sources of heavy metal contamination, including aspects of mine tailings composition and their environmental impact. This addition aims to strengthen the connection between the different components of the introduction.
- L76: Done.
Furthermore, we have incorporated additional details about mine tailings composition and their environmental impact to enhance the overall coherence and fluidity of the introduction. These modifications contribute to a clearer understanding of the relationship between environmental pollution, mine tailings, and heavy metal contamination. (L55-73)
- Results: In general. this section is well described. Please. you can find some specific comments to improve the section:
- L133: add the dot.
- L130: I suggest specifying the choice to use these two yeast strains. Which were the criteria used? Did you choose them focusing on particular PGP traits? If yes. which ones? Are there ordered criteria. depending also on the soil characteristics. that you follow? Specify it.
- L153: Specify the name of the mycorrhiza Claroideoglomus claroideum before inserting the abbreviation in brackets (CC)
- L194: Correct “percentage” with the case letter
A: Thank you for your thoughtful review of the Results section. We have addressed the specific comments provided and made the following modifications:
- L133: Done.
- L130: We have specified the criteria used for choosing the two yeast strains. The selection was based on particular Plant Growth-Promoting (PGP) traits. These criteria are detailed in the revised Results section (L140-142)
- L153: Done.
- L194: Done.
- Material and methods: The section is well structured and the experimental design is well described. However. my main concern is related to the analysis of the PGP traits. As reported in the manuscript. you use only qualitative methods to detect phosphate solubilization. ACC deaminase and siderophore production. These methods may be insufficient and misleading unless it is correlated with quantitative methods. I suggest further investigation to repeat the characterization of PGP traits using quantitative analysis to have a more precise estimation of the results.
A: We recognize that, in a previous revision of the article, the quantitative measurement of phosphate solubilization may not have been sufficiently clarified. We apologize for any misunderstanding and appreciate the opportunity to provide this clarification.
We want to clarify that phosphate solubilization was indeed quantitatively assessed, as detailed in Table 1, Material and Methods (554-562), and Results (L129-130) sections.
In addition to the quantitative measurement of phosphate solubilization, it is important to note that we also conducted quantitative analyses for Indole-3-Acetic Acid (IAA) production and Exopolysaccharide (EPS) content. These quantitative assessments, coupled with qualitative evaluations of siderophore production and ACC-deaminase activity, collectively contribute to a more comprehensive understanding of the Plant Growth-Promoting (PGP) potential exhibited by the selected yeast strains. We appreciate your insightful comments and are committed to further refining our methodologies in future investigations.
- The following elements need to be revised:
- L528-533: specify if you use sterile pots and soil during the in planta experiments.
- L531: add more information (like isolation sources etc...) about the AMF inoculum that you use
- L535: specify the choice to do the inoculation after 15 days. providing a reference.
- L536: correct 1x10-6 to 1x106
- L548: replace “replicated” with a more specific term (like purified)
- L639: specify which kind of modification you did. compared to the protocol mentioned above
- L657-663: specify which kind of modification you did. compared to the protocol mentioned above
A: Thank you for your meticulous review of the Methodology section. We have implemented the following corrections
- L528-533: We have clarified whether sterile pots (L533). Mine tailings were kept in their natural state (non-sterilized) to accurately simulate real-life conditions.
- L531: Additional information, such as isolation sources, has been included about the AMF inoculum used (L517-542).
- L535: This section was rewritten for greater clarity. A reference for the inoculation of yeasts was added (L524-525), and a reinoculation was performed at the time of transplanting (L533-537)
- L536: The notation "1x10-6" has been corrected to "1x10^6."
- L548: The term "replicated" has been replaced with "purified."
- L639 and L657-663: We have revised this section of the methodology. Additionally, references have been added to provide support.
- Discussion: In general the discussion section is well organized. However. you should revise some points. as follows:
- L392: correct the measure unit expression (mg kg-1)
- L426-431: I suggest moving this part in the conclusion section as a future perspective. as a base for further investigation.
- L446: change the comma with the dot in decimal numbers
A: We have implemented the following changes in response to your feedback:
- L392: Done.
- L426-431: We retained the discussion on competitive relationships (L493-500), and further addressed this aspect in the conclusion as a perspective for future studies (L727-734)
- L446: Done.
- Conclusions: The authors presented a summary of the study. I suggest giving more details on the future research directions. Please. find below a specific comment:
- L735: "M. guilliermondii" should be in italics
A: As previously indicated, we have made the necessary revisions, incorporating more details on future research directions
- L735: Done.
- Other comments: All the sections need improvements for the English language.
A: Regarding to the language improvement, we would like to inform you that we subjected our manuscript to the scrutiny of a certified English language reviewer. We ensured to address and incorporate all suggested corrections to enhance the linguistic quality of our manuscript.
Round 2
Reviewer 2 Report
Comments and Suggestions for Authors
The authors have revised their manuscript. This version can be accepted.
Fig. 1D: correct "lenght" as "length"
Reviewer 3 Report
Comments and Suggestions for Authors
Dear authors,
the manuscript plants-2695358 entitled “Beneficial interactive effect provided by an arbuscular mycorrhizal fungi and yeast in the growth of Oenothera picensis established on Cu mine tailings” has now the potential to be published after the revisions that you have done.